# PROVABLY CONVERGENT NONCONVEX ALGORITHM FOR VOLUME OPTIMIZATION-BASED LATENT COMPONENT ANALYSES

## ABSTRACT

We present an algorithm that aims at solving a family of nonconvex optimization problems with convergence guarantee to a global optimum. This family of nonconvex optimization problems share the formulation of maximizing the volume of a matrix subject to linear constraints. Problems of this form have found a lot of applications in unsupervised learning and representation learning, especially when identifiability of the latent representation is important for the task. Specific examples based on the types of constraints include bounded component analysis, sparse component analysis (complete dictionary learning), nonnegative component analysis (nonnegative matrix factorization), and admixture component analysis, to name a few. Computationally, the problem is hard because of the nonconvex objective. An algorithm based on linearized ADMM is proposed for these problems. Although a similar algorithm has appeared in the literature, we note that a small modification has to be made in order to guarantee that the algorithm provably converges even for non-convex problems. Our main contribution is convergence guarantee to a global optimum at a sublinear rate. We do assume some mild conditions on the initialization, but our numerical experiments indicate that these initialization conditions are very easy to satisfy.

## 1 INTRODUCTION

Many problems in unsupervised representation learning require identifying the latent components from data with structural assumptions/constraints on the latent representation. While some works show that volume optimization can guarantee identifiability of the latent components under mild conditions, the provably convergent algorithm is still missing for any of these problems. To bridge this gap, we propose an algorithm that aims at solving the following (family of) nonconvex optimization problem:

$$\underset{\boldsymbol{W}}{\text{minimize}} \quad -\frac{1}{2}\log\det\boldsymbol{W}\boldsymbol{W}^{\top} + g(\boldsymbol{W}\boldsymbol{X}), \tag{1}$$

where $g$ is some regularization function to enforce or promote certain problem-specific structure to the transformation $\boldsymbol{W}\boldsymbol{X}$ to a lower dimension, therefore $\boldsymbol{W}$ should be a wide matrix with full row rank. This type of problems naturally arises in many unsupervised learning problems, where data samples are stacked in the columns of $\boldsymbol{X}$ and the goal is to find a linear transformation $\boldsymbol{W}$ that makes the learned representation/embedding $\boldsymbol{W}\boldsymbol{X}$ satisfy some structural requirements. A classical example is independent component analysis (ICA), where ideally $g(\boldsymbol{W}\boldsymbol{X})$ should be a function to enforce that the rows of $\boldsymbol{W}\boldsymbol{X}$ are statistically independent (Comon, 1994; Hyvärinen et al., 2004), although in practice it is fairly hard to find such a well-defined function, especially if working with a finite set of data $\boldsymbol{X}$. Ever since, some structural constraints/regularizations that are more suitable for finite data sets with explicitly well-defined functional forms have been studied, including:

- Bounded component analysis (BCA) (Tatli & Erdogan, 2021; Hu & Huang, 2023b). In this case $g$ is an indicator function defined as

$$g(\boldsymbol{S}) = \begin{cases} 0 & |S_{ij}| \le 1 \text{ for all } i,j \\ +\infty & \text{otherwise.} \end{cases}$$

Notice that $|\cdot|$ could mean magnitude of a complex number if data are complex, in which case replace matrix transpose $^\top$ with Hermitian transpose $^\mathsf{H}$ (Hu & Huang, 2024).

- Sparse component analysis (SCA) (Hu & Huang, 2023a), also known as (complete) dictionary learning. Here $g$ is simply the (vector) $\ell_1$ norm, i.e., sum of absolute values

$$g(\boldsymbol{S}) = \|\boldsymbol{S}\|_1 = \sum_{i,j} |S_{ij}|.$$

Again, it can be extended to complex numbers with ease (Sun & Huang, 2024). We will not consider overcomplete dictionaries in this paper, although a related formulation has appeared (Sun & Huang, 2025).

- Nonnegative component analysis (NCA) (Fu et al., 2018), which is a variant of nonnegative matrix factorization (NMF) with only one of the factors to be nonnegative:

$$g(\boldsymbol{S}) = \begin{cases} \sum_{i,j} S_{ij} & S_{ij} \geq 0 \text{ for all } i,j \\ +\infty & \text{otherwise.} \end{cases}$$

- Admixture component analysis (ACA). In this case, each embedding $\boldsymbol{W}\boldsymbol{x}_i$ represent the abundance/admixture of the components, meaning the coefficients are nonnegative and sum to one, so $g$ is again an indicator function

$$g(\boldsymbol{S}) = \begin{cases} 0 & \boldsymbol{S} \geq 0, \boldsymbol{I}^\top \boldsymbol{S} = \boldsymbol{I}^\top \\ +\infty & \text{otherwise.} \end{cases}$$

ACA has been widely adopted in topic models (Blei, 2012) and hyperspectral unmixing (Ma et al., 2013).

- Principal component analysis (PCA). We throw in PCA here for completeness, even though it is unnecessary as there are countless algorithms that efficiently solves it. We show in the supplementary that if $g(\boldsymbol{S}) = \|\boldsymbol{S}\|_\mathrm{F}^2$, the solution of (1) recovers that of PCA, exhibiting versatility of the framework.

## 1.1 Identifiability via Volume Optimization

For a wide matrix $\boldsymbol{W}$ with linearly independent rows, $\sqrt{\det \boldsymbol{W}\boldsymbol{W}^\top}$ is sometimes called the volume of the matrix (Ben-Israel, 1992) with applications in change-of-variable integration (Ben-Israel, 1999) and probability (Ben-Israel, 2000). In particular, if $\boldsymbol{s} \in \mathbb{R}^k$ is a latent random variable with distribution $p(\boldsymbol{s})$, and we observe $\boldsymbol{x} = \boldsymbol{A}\boldsymbol{s}$, then the distribution of $\boldsymbol{x}$ admits the following relation

$$p(\boldsymbol{x}) = (\det \boldsymbol{A}^\top \boldsymbol{A})^{-1/2} p(\boldsymbol{s}).$$

Therefore, applying a change-of-variable $\boldsymbol{A} = \boldsymbol{W}^\dagger$, formulation (1) is exactly the maximum likelihood estimate with the following latent distributions:

- PCA: standard Gaussian distribution
- SCA: standard Laplacian distribution
- NCA: standard exponential distribution
- BCA: uniform distribution in the hypercube
- ACA: uniform Dirichlet distribution

Even for general ICA models, as long as the type of latent variables are known, one can write down the exact maximum likelihood formulation in the form of (1), for instance the Cauchy distribution with $g(s) = -\log(1 + s^2)$.

The bigger benefit is identifiability, with the exception of PCA. Suppose the observations are indeed generated from the model $\boldsymbol{X} = \boldsymbol{A}\boldsymbol{S}$ with unknown $\boldsymbol{A}$ and $\boldsymbol{S}$, in many cases one is interested in exactly recover $\boldsymbol{S}$ up to trivial ambiguities such as row permutation and scaling. Historically, the original motivation to propose ICA was exactly because PCA is not identifiable. However, exact identifiability of ICA requires statistical independence, which is not clear how to quantify with finite data. The volume optimization framework for structured latent component analysis has been shown to guarantee identifiability under realistic assumptions. In particular, denote the true latent components as $\boldsymbol{S}_\natural$ and a solution to (1) as $\boldsymbol{S}_\star = \boldsymbol{W}_\star \boldsymbol{X}$, then $\boldsymbol{S}_\star$ is a row permutation and scaling of $\boldsymbol{S}_\natural$ if

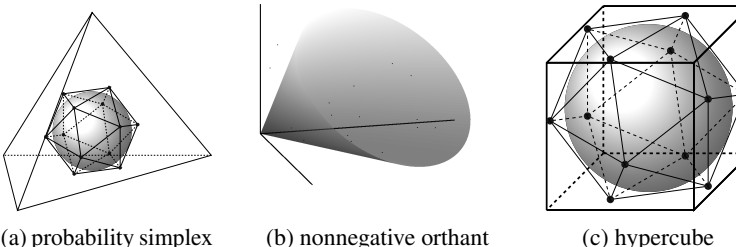

(a) probability simplex     (b) nonnegative orthant     (c) hypercube

Figure 1: Illustrations of the sufficiently scattered condition in the (a) probability simplex, (b) nonnegative orthant, and (c) hypercube in $\mathbb{R}^3$.

- ACA: convex hull of $\boldsymbol{S}_\natural$, denoted as $\mathrm{conv}(\boldsymbol{S}_\natural) = \{\boldsymbol{S}_\natural\boldsymbol{\theta} \mid \boldsymbol{\theta} \geq 0, \boldsymbol{I}^\top\boldsymbol{\theta} = 1\}$, is sufficiently scattered in the probability simplex (Fu et al., 2015; Lin et al., 2015)

$$\{\boldsymbol{x} \in \mathbb{R}^k \mid \|\boldsymbol{x}\|_2 \leq 1/\sqrt{k-1}, \boldsymbol{I}^\top\boldsymbol{x} = 1\} \subset \mathrm{conv}(\boldsymbol{S}_\natural).$$

- NCA: conic hull of $\boldsymbol{S}_\natural$, denoted as $\mathrm{cone}(\boldsymbol{S}_\natural) = \{\boldsymbol{S}_\natural\boldsymbol{\theta} \mid \boldsymbol{\theta} \geq 0\}$, is sufficiently scattered in the nonnegative orthant (Huang et al., 2013; Fu et al., 2018)

$$\{\boldsymbol{x} \in \mathbb{R}^k \mid \|\boldsymbol{x}\|_2 \leq \boldsymbol{I}^\top\boldsymbol{x}/\sqrt{k-1}\} \subset \mathrm{cone}(\boldsymbol{S}_\natural).$$

- BCA: disked hull of $\boldsymbol{S}_\natural$, denoted as $\mathrm{disk}(\boldsymbol{S}_\natural) = \{\boldsymbol{S}_\natural\boldsymbol{\theta} \mid \|\boldsymbol{\theta}\|_1 \leq 1\}$, is sufficiently scattered in the hypercube (Tatli & Erdogan, 2021; Hu & Huang, 2023b)

$$\{\boldsymbol{x} \in \mathbb{R}^k \mid \|\boldsymbol{x}\|_2 \leq 1\} \subset \mathrm{disk}(\boldsymbol{S}_\natural).$$

- SCA: cellular hull of $\tilde{\boldsymbol{S}}_\natural$, where $\tilde{\boldsymbol{S}}_\natural$ is obtained from rescaling the rows of $\boldsymbol{S}_\natural$ to unit $\ell_1$ norm, denoted as $\mathrm{disk}(\tilde{\boldsymbol{S}}_\natural) = \{\tilde{\boldsymbol{S}}_\natural\boldsymbol{\theta} \mid \|\boldsymbol{\theta}\|_\infty \leq 1\}$, is sufficiently scattered in the hypercube (Hu & Huang, 2023a)

$$\{\boldsymbol{x} \in \mathbb{R}^k \mid \|\boldsymbol{x}\|_2 \leq 1\} \subset \mathrm{cell}(\tilde{\boldsymbol{S}}_\natural).$$

Notice that the complex case for BCA (Hu & Huang, 2024) and SCA (Sun & Huang, 2024) are also true.

Illustrations of these various sufficiently scattered conditions are shown in Figure 1. More details on identifiability of these models can be found in the aforementioned references and therein.

## 1.2 THIS PAPER

This work focuses on algorithmic analysis for (1). First we note that (1) is nonconvex due to the volume term $-\log\det\boldsymbol{W}\boldsymbol{W}^\top$; although log-determinant of a positive definite matrix is concave, it is not the case when the positive definite matrix is parameterized by $\boldsymbol{W}\boldsymbol{W}^\top$. On the other hand, $g$ is usually a convex function or indicator function of a convex set, and usually admits an efficient proximal/projection operator. Some prior work on algorithmic design for (1) are reviewed in the supplementary. In this paper we propose a new algorithm based on the linearized alternating direction method of multipliers (L-ADMM) with a constant step size $\gamma$ as follows

$$\begin{cases} \boldsymbol{W}_{t+1} \leftarrow (\boldsymbol{S}_t + \boldsymbol{U}_t + \gamma\boldsymbol{S}_t^{\dagger\top})\boldsymbol{X}^\dagger \\ \boldsymbol{S}_{t+1} \leftarrow \mathrm{Prox}_{\gamma g}(\boldsymbol{W}_{t+1}\boldsymbol{X} - \boldsymbol{U}_t) \\ \boldsymbol{U}_{t+1} \leftarrow \boldsymbol{U}_t + \boldsymbol{S}_{t+1} - \boldsymbol{W}_{t+1}\boldsymbol{X} \end{cases} . \tag{2}$$

Derivation of the algorithm will be explained in detail in the sequel. An immediate observation is that (2) is very easy to implement in practice, as it only involves basic operations such as matrix multiplication, pseudo-inverse, and proximal operators—for the aforementioned specific examples, their corresponding proximal operators are as simple as truncations (for BCA) or soft-thresholding (for SCA). Despite its simplicity, its performance is surprisingly good, as will be demonstrated in the experiment section. Recognizing that we are trying to solve a nonconvex problem (1), with some

cases shown to be NP-hard (Packer, 2002), those numerical results seem incredible as it almost always achieves global optimality within a short amount of time. This inspires us to further investigate the algorithmic behavior of (2) to explain its excellent performance. In this paper we make the following contributions:

1. We proposed a general volume-optimization framework that includes the aforementioned problems (e.g., BCA/SCA/NCA/ACA, which are generally non-convex and NP-hard). More importantly, our newly designed algorithm is the first method with provable convergence to global optimum.

2. Compared to state-of-the-art baselines (BCD, FW, ADMM algorithms) that tailor designed for each identifiable problems, our new method is either significantly faster or at least comparable. This is also a non-trivial contribution on top of our theoretical results.

We want to remark our major technical significances. We introduce one version of linearized alternating direction method of multipliers (L-ADMM) that has not been proposed to the best of our knowledge. The reason we choose this version is thanks to its simplicity in proving convergence for convex problems, with minimal assumptions. It inspired us to generalize into non-convex problem.

Recognizing that convexity of the objective function is only used to form first-order lowerbounds and the other point is always a global optimum, a similar inequality is established for (2) even though $-\log \det \boldsymbol{W}\boldsymbol{W}^\top$ is not convex. The claim is that if $\boldsymbol{X}$ is indeed generated from an identifiable model $\boldsymbol{X} = \boldsymbol{A}_\natural \boldsymbol{S}_\natural$ and the initialization $\boldsymbol{S}_0$ satisfies

$$\log |\det \boldsymbol{S}_0 \boldsymbol{S}_\natural^\dagger| \le \max_{\boldsymbol{\Pi}, \boldsymbol{D}} \operatorname{Tr} \boldsymbol{\Pi} \boldsymbol{D} \boldsymbol{S}_0 \boldsymbol{S}_\natural^\dagger - k, \tag{3}$$

where $\boldsymbol{\Pi}$ are permutation matrices and $\boldsymbol{D}$ are diagonal matrices with $\pm 1$ on the diagonal, then L-ADMM converges to a global optimum. Although we do not provide an initialization scheme that always satisfies (3), our numerical experiments suggest that it is very easy to satisfy (3) with random initializations, thus explaining the above mentioned excellent performance.

## 2 LINEARIZED ADMM

Alternating direction method of multipliers (ADMM) has been a widely adopted algorithm in machine learning (Boyd et al., 2011). Consider a generic linearly constrained optimization problem

$$\begin{aligned} \underset{\boldsymbol{w}, \boldsymbol{s}}{\text{minimize}} \quad & f(\boldsymbol{w}) + g(\boldsymbol{s}) \\ \text{subject to} \quad & \boldsymbol{A}\boldsymbol{w} + \boldsymbol{B}\boldsymbol{s} = \boldsymbol{c}, \end{aligned} \tag{4}$$

ADMM takes the following iterative form

$$\begin{cases} \boldsymbol{w}_{t+1} \leftarrow \arg\min_{\boldsymbol{w}} f(\boldsymbol{w}) + \frac{1}{2\gamma}\|\boldsymbol{A}\boldsymbol{w} + \boldsymbol{B}\boldsymbol{s}_t - \boldsymbol{c} + \boldsymbol{u}_t\|^2 \\ \boldsymbol{s}_{t+1} \leftarrow \arg\min_{\boldsymbol{s}} g(\boldsymbol{s}) + \frac{1}{2\gamma}\|\boldsymbol{A}\boldsymbol{w}_{t+1} + \boldsymbol{B}\boldsymbol{s} - \boldsymbol{c} + \boldsymbol{u}_t\|^2 \\ \boldsymbol{u}_{t+1} \leftarrow \boldsymbol{u}_t + \boldsymbol{A}\boldsymbol{w}_{t+1} + \boldsymbol{B}\boldsymbol{s}_{t+1} - \boldsymbol{c} \end{cases}$$

where $\gamma$ is the step size. One of the possible downside is that it involves potentially nontrivial minimizations within each iteration that may not admit to closed-form updates. Suppose this is true for the update of $\boldsymbol{w}$, then one way to mitigate this is to linearize $f(\boldsymbol{w})$ in each iteration, leading to Linearized ADMM (L-ADMM)

$$\begin{cases} \boldsymbol{w}_{t+1} \leftarrow \arg\min_{\boldsymbol{w}} \nabla f(\boldsymbol{A}^\dagger(\boldsymbol{c} - \boldsymbol{B}\boldsymbol{s}_t))^\top \boldsymbol{w} + \frac{1}{2\gamma}\|\boldsymbol{A}\boldsymbol{w} + \boldsymbol{B}\boldsymbol{s}_t - \boldsymbol{c} + \boldsymbol{u}_t\|^2 \\ \boldsymbol{s}_{t+1} \leftarrow \arg\min_{\boldsymbol{s}} g(\boldsymbol{s}) + \frac{1}{2\gamma}\|\boldsymbol{A}\boldsymbol{w}_{t+1} + \boldsymbol{B}\boldsymbol{s} - \boldsymbol{c} + \boldsymbol{u}_t\|^2 \\ \boldsymbol{u}_{t+1} \leftarrow \boldsymbol{u}_t + \boldsymbol{A}\boldsymbol{w}_{t+1} + \boldsymbol{B}\boldsymbol{s}_{t+1} - \boldsymbol{c} \end{cases} \tag{5}$$

Note that the first order Taylor approximation of $f$ is taken at $\boldsymbol{A}^\dagger(\boldsymbol{c} - \boldsymbol{B}\boldsymbol{s}_t)$, not $\boldsymbol{w}_t$ as in (Hu & Huang, 2023a). To the best of our knowledge, this type of linearization has not appeared in the literature, among the various inexact ADMM variants (He et al., 2002; Ng et al., 2011; Lin et al., 2017; Gao et al., 2018; Lu et al., 2021). However, we find that this version leads to a relatively simple convergence proof for convex problems with minimal assumptions, i.e., $f$ is Lipschitz smooth and

columns of $A$ are linearly independent. This would be a particularly useful property for us to further apply it to nonconvex problems that are of interest in this work, as we will see shortly.

Here we provide a concise convergence proof for L-ADMM assuming both $f$ and $g$ are convex, while $f$ is also Lipschitz smooth. Our analysis differs from most existing work by employing a different Lyapunov function, which will be shown to converge to zero. Consider the Lagrangian function of (4)

$$L(\boldsymbol{w}, \boldsymbol{s}, \boldsymbol{\lambda}) = f(\boldsymbol{w}) + g(\boldsymbol{s}) + \boldsymbol{\lambda}^\top (A\boldsymbol{w} + B\boldsymbol{s} - \boldsymbol{c}).$$

If strong duality holds, then we have

$$L(\boldsymbol{w}, \boldsymbol{s}, \boldsymbol{\lambda}_\star) \geq L(\boldsymbol{w}_\star, \boldsymbol{s}_\star, \boldsymbol{\lambda}_\star) = f(\boldsymbol{w}_\star) + g(\boldsymbol{s}_\star),$$

where $(\boldsymbol{w}_\star, \boldsymbol{s}_\star)$ and $\boldsymbol{\lambda}_\star$ are optimal primal and dual variables, since $(\boldsymbol{w}_\star, \boldsymbol{s}_\star)$ minimizes $L(\boldsymbol{w}, \boldsymbol{s}, \boldsymbol{\lambda}_\star)$ and they are also feasible, i.e.,

$$A\boldsymbol{w}_\star + B\boldsymbol{s}_\star = \boldsymbol{c}. \tag{6}$$

We will show in the sequel that $L(\boldsymbol{w}_t, \boldsymbol{s}_t, \boldsymbol{\lambda}_\star) - L(\boldsymbol{w}_\star, \boldsymbol{s}_\star, \boldsymbol{\lambda}_\star)$, which is a nonnegative sequence, goes to zero as $t \to \infty$.

**Theorem 2.1.** *Suppose $f$ and $g$ are convex functions, $f$ is Lipschitz smooth, meaning there exists a constant $M$ such that for all $\boldsymbol{w}$ and $\tilde{\boldsymbol{w}}$ in the domain of $f$*

$$\|\nabla f(\boldsymbol{w}) - \nabla f(\tilde{\boldsymbol{w}})\| \leq M\|\boldsymbol{w} - \tilde{\boldsymbol{w}}\|,$$

*and matrix $A$ has full column rank: there exists a constant $\mu$ such that*

$$A^\top A \succeq \mu I.$$

*Then with $\gamma \leq \mu/M$, we have*

$$\min_{t=1,\dots,T} L(\boldsymbol{w}_t, \boldsymbol{s}_t, \boldsymbol{\lambda}_\star) - L(\boldsymbol{w}_\star, \boldsymbol{s}_\star, \boldsymbol{\lambda}_\star) \leq \frac{1}{T} \left( \frac{1}{2\gamma} \|A(\boldsymbol{w}_0 - \boldsymbol{w}_\star) + \boldsymbol{u}_0 - \gamma \boldsymbol{\lambda}_\star\|^2 \right). \tag{7}$$

*Proof.* First we rearrange the order of L-ADMM to

$$\begin{cases} \boldsymbol{s}_{t+1} \leftarrow \arg\min_{\boldsymbol{s}} g(\boldsymbol{s}) + \frac{1}{2\gamma}\|A\boldsymbol{w}_t + B\boldsymbol{s} - \boldsymbol{c} + \boldsymbol{u}_t\|^2 \\ \boldsymbol{u}_{t+1} \leftarrow \boldsymbol{u}_t + A\boldsymbol{w}_t + B\boldsymbol{s}_{t+1} - \boldsymbol{c} \\ \boldsymbol{w}_{t+1} \leftarrow \arg\min_{\boldsymbol{w}} \nabla f(A^\dagger(\boldsymbol{c} - B\boldsymbol{s}_{t+1}))^\top \boldsymbol{w} + \frac{1}{2\gamma}\|A\boldsymbol{w} + B\boldsymbol{s}_{t+1} - \boldsymbol{c} + \boldsymbol{u}_{t+1}\|^2 \end{cases} \tag{8}$$

This is equivalent to decreasing the iteration index of $\boldsymbol{w}$ by 1.

The update of $\boldsymbol{s}$ implies

$$-\frac{1}{\gamma} B^\top (A\boldsymbol{w}_t + B\boldsymbol{s}_{t+1} - c + \boldsymbol{u}_t) \in \partial g(\boldsymbol{s}_{t+1}),$$

for convex $g$ we have

$$g(\boldsymbol{s}_{t+1}) - g(\boldsymbol{s}_\star) \leq \frac{1}{\gamma}(A\boldsymbol{w}_t + B\boldsymbol{s}_{t+1} - \boldsymbol{c} + \boldsymbol{u}_t)^\top B(\boldsymbol{s}_\star - \boldsymbol{s}_{t+1}). \tag{9}$$

The update of $\boldsymbol{u}_{t+1}$ in (5) implies

$$B\boldsymbol{s}_{t+1} = \boldsymbol{c} - A\boldsymbol{w}_t + \boldsymbol{u}_{t+1} - \boldsymbol{u}_t.$$

Substituting it into (9) together with $B\boldsymbol{s}_\star = \boldsymbol{c} - A\boldsymbol{w}_\star$ from (6) gives

$$g(\boldsymbol{s}_{t+1}) - g(\boldsymbol{s}_\star) \leq \frac{1}{\gamma} \boldsymbol{u}_{t+1}^\top (A\boldsymbol{w}_t - A\boldsymbol{w}_\star + \boldsymbol{u}_t - \boldsymbol{u}_{t+1}). \tag{10}$$

Denote $\hat{\boldsymbol{w}}_t = A^\dagger(\boldsymbol{c} - B\boldsymbol{s}_{t+1})$. From convexity and Lipschitz-smoothness of $f$, we have

$$f(\boldsymbol{w}_{t+1}) \leq f(\hat{\boldsymbol{w}}_t) + \nabla f(\hat{\boldsymbol{w}}_t)^\top (\boldsymbol{w}_{t+1} - \hat{\boldsymbol{w}}_t) + \frac{M}{2}\|\boldsymbol{w}_{t+1} - \hat{\boldsymbol{w}}_t\|^2,$$
$$f(\boldsymbol{w}_\star) \geq f(\hat{\boldsymbol{w}}_t) + \nabla f(\hat{\boldsymbol{w}}_t)^\top (\boldsymbol{w}_\star - \hat{\boldsymbol{w}}_t). \tag{11}$$

From the update of $\boldsymbol{w}$ in (5), we have

$$\nabla f(\hat{\boldsymbol{w}}_t) = -\frac{1}{\gamma}\boldsymbol{A}^\top(\boldsymbol{A}\boldsymbol{w}_{t+1} + \boldsymbol{B}\boldsymbol{s}_{t+1} - \boldsymbol{c} + \boldsymbol{u}_{t+1}) = -\frac{1}{\gamma}\boldsymbol{A}^\top(\boldsymbol{A}\boldsymbol{w}_{t+1} - \boldsymbol{A}\boldsymbol{w}_t + 2\boldsymbol{u}_{t+1} - \boldsymbol{u}_t)$$

All three equations above shows

$$f(\boldsymbol{w}_{t+1}) - f(\boldsymbol{w}_\star) \le \frac{1}{\gamma}(\boldsymbol{A}\boldsymbol{w}_{t+1} - \boldsymbol{A}\boldsymbol{w}_t + 2\boldsymbol{u}_{t+1} - \boldsymbol{u}_t)^\top \boldsymbol{A}(\boldsymbol{w}_\star - \boldsymbol{w}_{t+1}) + \frac{M}{2}\|\boldsymbol{w}_{t+1} - \hat{\boldsymbol{w}}_t\|^2. \quad (12)$$

Combining (10) and (12) gives

$$f(\boldsymbol{w}_{t+1}) + g(\boldsymbol{s}_{t+1}) - f(\boldsymbol{w}_\star) - g(\boldsymbol{s}_\star)$$

$$\le \frac{1}{\gamma}(\boldsymbol{A}\boldsymbol{w}_{t+1} - \boldsymbol{A}\boldsymbol{w}_t + 2\boldsymbol{u}_{t+1} - \boldsymbol{u}_t)^\top \boldsymbol{A}(\boldsymbol{w}_\star - \boldsymbol{w}_{t+1}) + \frac{1}{\gamma}\boldsymbol{u}_{t+1}^\top(\boldsymbol{A}\boldsymbol{w}_t - \boldsymbol{A}\boldsymbol{w}_\star + \boldsymbol{u}_t - \boldsymbol{u}_{t+1}) + \frac{M}{2}\|\boldsymbol{w}_{t+1} - \hat{\boldsymbol{w}}_t\|^2$$

$$= \frac{1}{\gamma}(\boldsymbol{A}\boldsymbol{w}_{t+1} - \boldsymbol{A}\boldsymbol{w}_t + \boldsymbol{u}_{t+1} - \boldsymbol{u}_t)^\top \boldsymbol{A}(\boldsymbol{w}_\star - \boldsymbol{w}_{t+1}) + \frac{1}{\gamma}\boldsymbol{u}_{t+1}^\top(\boldsymbol{A}\boldsymbol{w}_t - \boldsymbol{A}\boldsymbol{w}_{t+1} + \boldsymbol{u}_t - \boldsymbol{u}_{t+1}) + \frac{M}{2}\|\boldsymbol{w}_{t+1} - \hat{\boldsymbol{w}}_t\|^2$$

$$= \frac{1}{\gamma}(\boldsymbol{A}\boldsymbol{w}_{t+1} - \boldsymbol{A}\boldsymbol{w}_t + \boldsymbol{u}_{t+1} - \boldsymbol{u}_t)^\top(\boldsymbol{A}\boldsymbol{w}_\star - \boldsymbol{A}\boldsymbol{w}_{t+1} - \boldsymbol{u}_{t+1}) + \frac{M}{2}\|\boldsymbol{w}_{t+1} - \hat{\boldsymbol{w}}_t\|^2$$

Adding both sides by

$$\boldsymbol{\lambda}_\star^\top(\boldsymbol{A}\boldsymbol{w}_{t+1} + \boldsymbol{B}\boldsymbol{s}_{t+1} - \boldsymbol{c}) = \boldsymbol{\lambda}_\star^\top(\boldsymbol{A}\boldsymbol{w}_{t+1} - \boldsymbol{A}\boldsymbol{w}_t + \boldsymbol{u}_{t+1} - \boldsymbol{u}_t),$$

where the right-hand-side is obtained from the update rule of $\boldsymbol{u}_{t+1}$, we get

$$L(\boldsymbol{w}_{t+1}, \boldsymbol{s}_{t+1}, \boldsymbol{\lambda}_\star) - L(\boldsymbol{w}_\star, \boldsymbol{s}_\star, \boldsymbol{\lambda}_\star)$$

$$\le \frac{1}{\gamma}(\boldsymbol{A}\boldsymbol{w}_{t+1} - \boldsymbol{A}\boldsymbol{w}_t + \boldsymbol{u}_{t+1} - \boldsymbol{u}_t)^\top(\boldsymbol{A}\boldsymbol{w}_\star - \boldsymbol{A}\boldsymbol{w}_{t+1} + \gamma\boldsymbol{\lambda}_\star - \boldsymbol{u}_{t+1}) + \frac{M}{2}\|\boldsymbol{w}_{t+1} - \hat{\boldsymbol{w}}_t\|^2$$

$$= \frac{1}{2\gamma}\|\boldsymbol{A}\boldsymbol{w}_\star - \boldsymbol{A}\boldsymbol{w}_t + \gamma\boldsymbol{\lambda}_\star - \boldsymbol{u}_t\|^2 - \frac{1}{2\gamma}\|\boldsymbol{A}\boldsymbol{w}_\star - \boldsymbol{A}\boldsymbol{w}_{t+1} + \gamma\boldsymbol{\lambda}_\star - \boldsymbol{u}_{t+1}\|^2 \qquad (13)$$

$$- \frac{1}{2\gamma}\|\boldsymbol{A}\boldsymbol{w}_{t+1} - \boldsymbol{A}\boldsymbol{w}_t + \boldsymbol{u}_{t+1} - \boldsymbol{u}_t\|^2 + \frac{M}{2}\|\boldsymbol{w}_{t+1} - \hat{\boldsymbol{w}}_t\|^2$$

Notice that we defined $\hat{\boldsymbol{w}}_t = \boldsymbol{A}^\dagger(\boldsymbol{c} - \boldsymbol{B}\boldsymbol{s}_{t+1}) = \boldsymbol{A}^\dagger(\boldsymbol{A}\boldsymbol{w}_t + \boldsymbol{u}_t - \boldsymbol{u}_{t+1})$, and if $\boldsymbol{A}^\top \boldsymbol{A} \succeq \mu\boldsymbol{I}$ and $\gamma \le \mu/M$, then

$$\frac{M}{2}\|\boldsymbol{w}_{t+1} - \hat{\boldsymbol{w}}_t\|^2 \le \frac{M}{2\mu}\|\boldsymbol{A}\boldsymbol{w}_{t+1} - \boldsymbol{A}\boldsymbol{w}_t + \boldsymbol{u}_{t+1} - \boldsymbol{u}_t\|^2 \le \frac{1}{2\gamma}\|\boldsymbol{A}\boldsymbol{w}_{t+1} - \boldsymbol{A}\boldsymbol{w}_t + \boldsymbol{u}_{t+1} - \boldsymbol{u}_t\|^2.$$

This means the last line of (13) is nonpositive, and thus

$$L(\boldsymbol{w}_{t+1}, \boldsymbol{s}_{t+1}, \boldsymbol{\lambda}_\star) - L(\boldsymbol{w}_\star, \boldsymbol{s}_\star, \boldsymbol{\lambda}_\star)$$

$$= \frac{1}{2\gamma}\|\boldsymbol{A}\boldsymbol{w}_\star - \boldsymbol{A}\boldsymbol{w}_t + \gamma\boldsymbol{\lambda}_\star - \boldsymbol{u}_t\|^2 - \frac{1}{2\gamma}\|\boldsymbol{A}\boldsymbol{w}_\star - \boldsymbol{A}\boldsymbol{w}_{t+1} + \gamma\boldsymbol{\lambda}_\star - \boldsymbol{u}_{t+1}\|^2 \qquad (14)$$

Taking the summation of (14) with $t = 1, \ldots, T$, omit the negative terms on the right-hand-side, and replace each $L(\boldsymbol{w}_t, \boldsymbol{s}_t, \boldsymbol{\lambda}_\star)$ with their min on the left-hand-side, gives us (7). □

The proof clearly explains why the linearization is taken at $\hat{\boldsymbol{w}}_t = \boldsymbol{A}^\dagger(\boldsymbol{c} - \boldsymbol{B}\boldsymbol{s}_{t+1})$: the Lipschitz-smoothness property introduces a nonnegative term $(M/2)\|\boldsymbol{w}_{t+1} - \hat{\boldsymbol{w}}_t\|^2$ that needs to be eliminated in order to guarantee convergence, and the last line of (13) provides the choice of $\hat{\boldsymbol{w}}_t$ to achieve this. Numerically, this result hints that one could precondition the matrix $\boldsymbol{A}$ for better numerical performance (and a larger range of choice for $\gamma$). We find that orthogonizing the columns of $\boldsymbol{A}$ works really well in practice; since the update of $\boldsymbol{w}$ involves solving a least squares problem, a linear transformation of $\boldsymbol{w}$ is not going to change the complexity of its update. The proof also suggests that the linearization step should be done first (or, equivalently, immediately after the dual update), not the other way around, otherwise convergence may not be guaranteed. We obtained a sublinear convergence rate of $1/T$ without assuming strong convexity, which is similar to proximal gradient descent. If we assume $f$ to be strongly convex, one may expect to achieve a faster linear convergence rate with ease, but that is beyond the scope of this work.

**Remark.** Throughout the proof, convexity of $f$ and $g$ are only invoked to form the linear lower-bounds (9) and (11); moreover, one of the points is always at the optimum $s_\star$ or $w_\star$. This observation hints that even if, say, $f$ is not convex, as long as the iterates of L-ADMM generates $\hat{w}_t$ that satisfies (11), then global optimality can still be guaranteed. This is going to be the main focus when applying L-ADMM to nonconvex problems.

## 3 L-ADMM FOR VOLUME OPTIMIZATION-BASED LATENT COMPONENT ANALYSES

Before applying L-ADMM, we first reformulate (1) by introducing auxiliary variable $S = WX$ as

$$\underset{W,S}{\text{minimize}} \quad -\frac{1}{2}\log\det WW^\top + g(S) \tag{15}$$
$$\text{subject to} \quad S = WX.$$

This formulation has two separate terms for $W$ and $S$ coupled by a linear constraint $S = WX$, which is a form that L-ADMM can easily be applied. Its exact form has been presented in (2).

### 3.1 IMPLEMENTATION DETAILS

The updates of $S$ and $U$ follows exactly from the definition of L-ADMM in (5). The update of $W$ requires a bit more explanation. First of all, the gradient of we know the gradient of the log-determinant objective is $-(W^\dagger)^\top$ and since it is evaluated at $S_t X^\dagger$, we have that

$$\nabla -\log\det(S_t X^\dagger)(S_t X^\dagger)^\top = -(S_t X^\dagger)^{\dagger\top} = -S_t^{\dagger\top} X^\top,$$

where the last equality uses the fact that $(AB)^\dagger = (A^\dagger AB)^\dagger (ABB^\dagger)^\dagger$ (Petersen et al., 2008), which further equals to $B^\dagger A^\dagger$ if $A$ has full column rank and $B$ has full row rank. The update of $W_{t+1}$ is explicitly defined as

$$W_{t+1} = \arg\min_W -\text{Tr}\, W^\top S_t^{\dagger\top} X^\top + \frac{1}{2\gamma}\|S_t - WX + U_t\|^2.$$

Setting its gradient equals zero gives the $W$ update in (2).

As for the proximal operators for various $g$, it has been studied extensively, so we simply list some of them here without derivations:

$$\text{BCA:} \quad \text{Prox}_{\gamma g}(S)_{ij} = \begin{cases} \text{sign}(S_{ij}), & |S_{ij}| > 1, \\ S_{ij}, & \text{otherwise.} \end{cases}$$

$$\text{SCA:} \quad \text{Prox}_{\gamma g}(S)_{ij} = \begin{cases} \text{sign}(S_{ij})(|S_{ij}| - \gamma), & |S_{ij}| > \gamma, \\ 0, & \text{otherwise.} \end{cases}$$

$$\text{NCA:} \quad \text{Prox}_{\gamma g}(S)_{ij} = [S_{ij} - \gamma]_+$$

$$\text{ACA:} \quad \text{Prox}_{\gamma g}(s_i) = [s_i - v_i I]_+$$

Simplex projection for ACA is taken column-wise of $S$ and each $v_i$ is a scalar to satisfy that $I^\top[s_i - v_i I]_+ = 1$; there exist several methods that keep the complexity linear, such as bisection (Parikh & Boyd, 2014) or divide-and-conquer (Duchi et al., 2008).

### 3.2 GLOBAL CONVERGENCE

We now present the main contribution of this work, namely to show that Algorithm (2) converges to a global optimum of (1) under some mild conditions. As formulation (1) is mostly used for identifiable latent component analyses such as BCA or SCA, we focus on the case when $X$ is generated from $X = A_\natural S_\natural$, where $A_\natural$ and $S_\natural$ are the true latent factors, and $S_\natural$ satisfies one of the sufficiently scattered conditions described in §1.1, therefore any solution to (1) must be a row permutation and sign flip (if BCA or SCA) of $S_\natural$.

Denote $f(\mathbf{W}) = -\log \det \mathbf{W}\mathbf{W}^{\top}$. Proof of Theorem 2.1 indicates that convexity of $f$ is only invoked when forming a linear lowerbound (11) at $\mathbf{S}_t \mathbf{X}^{\dagger} = \mathbf{S}_t(\mathbf{A}_{\natural}\mathbf{S}_{\natural})^{\dagger}$ and $\mathbf{W}_{\star} = \mathbf{A}_{\natural}^{\dagger}$, i.e.,

$$-\frac{1}{2}\log\det(\mathbf{A}_{\natural}^{\dagger}\mathbf{A}_{\natural}^{\dagger\top}) \geq -\frac{1}{2}\log\det(\mathbf{S}_t(\mathbf{A}_{\natural}\mathbf{S}_{\natural})^{\dagger}(\mathbf{A}_{\natural}\mathbf{S}_{\natural})^{\dagger\top}\mathbf{S}_t^{\top}) - \operatorname{Tr}\mathbf{X}\mathbf{S}_t^{\top}(\mathbf{A}_{\natural}^{\dagger} - \mathbf{S}_t\mathbf{X}^{\dagger}).$$

This simplifies to

$$\log|\det \mathbf{S}_t\mathbf{S}_{\natural}^{\dagger}| \leq \operatorname{Tr}\mathbf{S}_t\mathbf{S}_{\natural}^{\dagger} - k. \tag{16}$$

We should be reminded that (16) does not always hold, but if it was true for all $\mathbf{S}_t$, then it suffices to guarantee that L-ADMM converges to $\mathbf{S}_{\natural}$. In fact, since any row permutation and sign flip (for BCA and SCA) of $\mathbf{S}_{\natural}$ is also optimal for (15), we can replace $\mathbf{S}_{\natural}$ in (16) with $\mathbf{\Pi}\mathbf{D}\mathbf{S}_{\natural}$ where $\mathbf{\Pi}$ is a permutation matrix and $\mathbf{D}$ is a diagonal matrix with $\pm 1$ on the diagonal (for NCA and ACA we would have $\mathbf{D} = \mathbf{I}$, but we include this term for completeness), then global convergence can be guaranteed.

To gain some insights about what (16) implies, obviously the inequality holds for positive definite matrices. However, $\mathbf{S}_t\mathbf{S}_{\natural}^{\dagger}$ is not positive definite. While the left-hand-side is related to the log-determinant of is Gram matrix, the right-hand-side only depends on its diagonal values. Therefore, intuitively, (16) would hold if $\mathbf{S}_t\mathbf{S}_{\natural}^{\dagger}$ is "diagonally dominant", to some extent. Indeed, if $\mathbf{S}_t$ is optimal, then $\mathbf{S}_t\mathbf{S}_{\natural}^{\dagger} = \mathbf{I}$, which is the most diagonally dominant matrix. To put it differently, (16) holds when $\mathbf{S}_t\mathbf{S}_{\natural}^{\dagger}$ is relatively close to $\mathbf{I}$; it does not have to be extremely close like many nonconvex algorithmic analysis requires, just close enough that (16) holds suffices. In other words, (16) defines the "basin of attraction" for the nonconvex problem (15).

**Theorem 3.1.** *Let $\mathbf{X} = \mathbf{A}_{\natural}\mathbf{S}_{\natural}$ where $\mathbf{A}_{\natural}$ and $\mathbf{S}_{\natural}$ are the true latent factors, and $\mathbf{S}_{\natural}$ satisfies one of the sufficiently scattered conditions described in §1.1. Suppose the L-ADMM algorithm (2) is initialized with $\mathbf{S}_0$ that satisfies*

$$\log|\det \mathbf{S}_0\mathbf{S}_{\natural}^{\dagger}| \leq \max_{\mathbf{\Pi},\mathbf{D}}\operatorname{Tr}\mathbf{\Pi}\mathbf{D}\mathbf{S}_0\mathbf{S}_{\natural}^{\dagger} - k,$$

*where $\mathbf{\Pi}$ is a permutation matrix and $\mathbf{D}$ is a diagonal matrix with $\pm 1$ on the diagonal, then we have*

$$\min_{t=1,\dots,T}L(\mathbf{W}_t, \mathbf{S}_t, \boldsymbol{\lambda}_{\star}) - L(\mathbf{W}_{\star}, \mathbf{S}_{\star}, \boldsymbol{\lambda}_{\star}) \leq \frac{1}{T}\left(\frac{1}{2\gamma}\|(\mathbf{W}_0\mathbf{X} - \mathbf{\Pi}\mathbf{S}_{\natural} + \mathbf{U}_0 - \gamma\boldsymbol{\lambda}_{\star}\|^2\right).$$

*Proof sketch.* The proof largely follows that of Theorem 2.1, except that in this case $f$ is not convex. However, we will show that as long as the initialization $\mathbf{S}_0$ satisfies

$$\log|\det \mathbf{S}_0\mathbf{S}_{\natural}^{\dagger}| \leq \max_{\mathbf{\Pi},\mathbf{D}}\operatorname{Tr}\mathbf{\Pi}\mathbf{D}\mathbf{S}_0\mathbf{S}_{\natural}^{\dagger} - k,$$

then (11) is satisfied throughout the iterations. All we need is to prove the following lemma. □

**Lemma 3.2.** *When running the L-ADMM iterations, if $\mathbf{S}_t$ satisfies*

$$\log|\det \mathbf{S}_t\mathbf{S}_{\natural}^{\dagger}| \leq \operatorname{Tr}\mathbf{\Pi}\mathbf{D}\mathbf{S}_t\mathbf{S}_{\natural}^{\dagger} - k,$$

*for some permutation matrix $\mathbf{\Pi}$ and diagonal matrix $\mathbf{D}$ with $\pm 1$ on the diagonal, then $\mathbf{S}_{t+1}$ also satisfies*

$$\log|\det \mathbf{S}_{t+1}\mathbf{S}_{\natural}^{\dagger}| \leq \operatorname{Tr}\mathbf{\Pi}\mathbf{D}\mathbf{S}_{t+1}\mathbf{S}_{\natural}^{\dagger} - k,$$

The proof is relegated to the supplementary.

## 4 EXPERIMENTS

We now provide some numerical experiments to showcase the effectiveness of the proposed L-ADMM for solving some special cases of (1), in particular BCA, SCA, NCA, and ACA. All the experiments are conducted in MATLAB on an iMac. We synthetically generate random problems. For $k = 20$ and $n = 1000$, we randomly generate the groundtruth factor matrices $\mathbf{S}^{\natural} \in \mathbb{R}^{k \times n}$ and $\mathbf{A}^{\natural} \in \mathbb{R}^{k \times k}$, and construct the data matrix $\mathbf{X} = \mathbf{A}^{\natural}\mathbf{S}^{\natural}$. Elements of $\mathbf{A}_{\natural}$ are independently drawn from a standard normal distribution, while those of $\mathbf{S}_{\natural}$ are generated as follows:

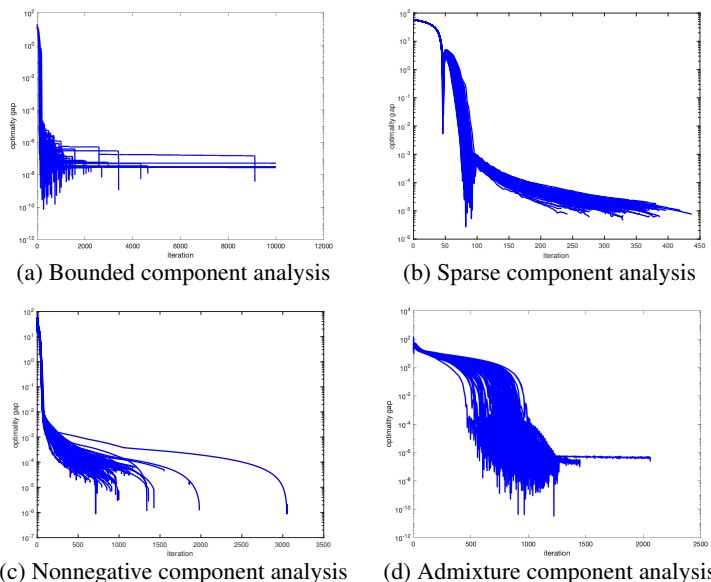

(a) Bounded component analysis      (b) Sparse component analysis

(c) Nonnegative component analysis      (d) Admixture component analysis

Figure 2: Convergence of L-ADMM in (2) for various latent component analyses models on 100 random trials.

- BCA: each $S_{ij}$ has 50% chance to be $\pm 1$ and 50% chance uniform in $[-1, 1]$.
- SCA: each $S_{ij}$ has 50% chance to be 0 and 50% chance standard normal; then each row is rescaled to have unit $\ell_1$ norm.
- NCA: each $S_{ij}$ has 50% chance to be 0 and 50% chance exponential; then each row is rescaled to sum to one.
- ACA: each $S_{ij}$ has 50% chance to be 0 and 50% chance exponential; then each column is rescaled to sum to one.

Prior works have shown that each of these generative models is identifiable with very high probability, despite $n$ being not that big compared to the number of atoms $k$. Matrix $\boldsymbol{X}$ is used as input to the L-ADMM algorithm as described in (2). Although Problem (15) is nonconvex, as long as it is identifiable, we know the global optimum is attained at $\boldsymbol{W}_\star == \boldsymbol{A}_\natural^\dagger$. As a result, $(1/2) \log \det \boldsymbol{A}_\natural^\top \boldsymbol{A}_\natural$ is the optimal value for Problem (15) as long as the model is identifiable, and we shall see whether the proposed algorithm is able to attain that optimal value. Inspired by the above convergence analysis, we check the optimality gap of the Lagrangian function values using the optimal dual variable $\boldsymbol{\Lambda}$

$$-(1/2) \log \det \boldsymbol{W}_t \boldsymbol{W}_t^\top + \mathrm{Tr}(\boldsymbol{S}_t - \boldsymbol{W}_t \boldsymbol{X}) \boldsymbol{\Lambda}_\star - (1/2) \log \det \boldsymbol{A}_\natural^\top \boldsymbol{A}_\natural.$$

Furthermore, it is easy to show that an optimal $\boldsymbol{\Lambda}$ is $\boldsymbol{S}_\natural^\dagger$. In this simulation with known groundtruth factors, we will use this to measure the optimality gap. The convergence behavior of 100 random trials of the L-ADMM are shown in Figure 2. Indeed, even though we are trying to solve a nonconvex problem (1), L-ADMM always converges to global optimum in our experiment. Each execution takes no more than a few seconds.

## 5 CONCLUSION

A general framework of volume optimization-based latent component analyses problems are studied, which includes many well-known unsupervised learning models such as dictionary learning, nonnegative matrix factorization, topic modeling, etc. An algorithm based on linearized ADMM (L-ADMM) is proposed, which admits simple update rules that are easy to implement in practice. Even though the problem is NP-hard, we show both in theory and in practice that the proposed algorithm is extremely effective at finding a global optimum.

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
