# PROVABLY CONVERGENT NONCONVEX ALGORITHM FOR VOLUME OPTIMIZATION-BASED LATENT COMPONENT ANALYSES

## SUPPLEMENTAL MATERIAL

## A    RELATED WORKS

In this section we introduce some existing algorithms that directly tackles the formulation, repeated here:

$$\underset{\boldsymbol{W}}{\text{minimize}} \quad -\frac{1}{2}\log\det \boldsymbol{W}\boldsymbol{W}^\top + g(\boldsymbol{W}\boldsymbol{X}). \tag{1}$$

More specifically, we introduce two algorithms that rely on existing off-the-shelf toolboxes to solve linear programming problems as a sub-routine in each iteration. There exists an early work based on augmented Lagrangian method specifically designed for ACA (Bioucas-Dias, 2009). We will include it in the experiment comparing all these algorithms, but the description is omitted as it is somewhat similar in the spirit of our proposed L-ADMM.

### A.1    BLOCK COORDINATE DESCENT (BCD)

This method only works when $\boldsymbol{W}$ is square, so $\det \boldsymbol{W}\boldsymbol{W}^\top = |\det \boldsymbol{W}|^2$. Using the Laplace's formula, we know that $\det \boldsymbol{W}$ is a linear function with respect to the $i$th row of $\boldsymbol{W}$ using the co-factor expansion

$$\det \boldsymbol{W} = \sum_{j=1}^{k}(-1)^{i+j}W_{ij}\det \boldsymbol{W}_{ij},$$

where the matrix $\boldsymbol{W}_{ij}$ is obtained by deleting the $i$th row and $j$th column of $\boldsymbol{W}$. Then to apply the block coordinate descent algorithm (Bertsekas, 1999), one would minimize $-|\boldsymbol{f}_i^\top\boldsymbol{w}_i| + g_i(\boldsymbol{w}_i^\top\boldsymbol{X})$, where $\boldsymbol{f}_i \in \mathbb{R}^k$ is a vector defined according to the co-factor expansion formula, and $g_i$ is a function in $\boldsymbol{R}^k$ by fixing all except the $i$th row of $\boldsymbol{W}$.

It looks as if we need to solve two linear programs for each row update, one to maximize $\boldsymbol{f}_i^\top\boldsymbol{w}_i$ and one to minimize $\boldsymbol{f}_i^\top\boldsymbol{w}_i$, but it turns out one only needs to maximize $\boldsymbol{f}_i^\top\boldsymbol{w}_i$. For BCA and SCA where there is a sign ambiguity in each component, either solution is a sign flip of the other and equally good; for NCA and ACA, due to the nonnegativity constraint, the sign of $\boldsymbol{f}_i^\top\boldsymbol{w}_i$ should equal to that of $\det \boldsymbol{W}$. Furthermore, recall that the Cramer's rule shows that

$$\left[\boldsymbol{P}^{-1}\right]_{ji} = (-1)^{i+j}\det \boldsymbol{P}_{ij}\Big/ \det \boldsymbol{P},$$

meaning we can simply define $\boldsymbol{f}_i$ as the $i$th column of $\boldsymbol{P}^{-1}$, and it would not affect the row updates.

The idea was first proposed to solve ACA by Chan et al. (2009). Several follow-up works have been proposed to solve related problems, such as NCA (Huang et al., 2016), ACA (Huang & Fu, 2019), BCA (Hu & Huang, 2023b), and SCA (Hu & Huang, 2023a). Notice that BCD works the best when the constraints are all separable over the blocks (Bertsekas, 1999); this makes BCD works fairly good on BCA, SCA, and NCA, but not particularly well on ACA (even though it was first proposed on this problem).

### A.2    FRANK-WOLFE (FW)

The Frank-Wolfe algorithm, also known as the conditional gradient method for constrained optimization (Bertsekas, 1999), iteratively minimizes a linear objective, defined by the gradient at the

---

**Algorithm 1** Solving (1) with BCD

---

initialize $\boldsymbol{W}_{(0)}$
**repeat**
  **for** $i = 1, \dots, k$ **do**
    $\boldsymbol{f} = \boldsymbol{W}^{-1}\boldsymbol{e}_i$
    $\boldsymbol{w} = \arg\min_{\boldsymbol{w}} -\boldsymbol{f}^\top\boldsymbol{w} + g_i(\boldsymbol{w}_i^\top\boldsymbol{X})$
    replace $i$th row of $\boldsymbol{W}$ with $\boldsymbol{w}^\top$
  **end for**
**until** convergence

---

current iterate, under the same constraint set to determine the search direction and obtain the next iterate via some line search approach along the search direction. For the log-determinant objective in (1), we have that the gradient is $-(\boldsymbol{P}^\dagger)^\top$. As a result, the Frank-Wolfe algorithm for (1) is given in Algorithm 2.

---

**Algorithm 2** Solving (1) with Frank-Wolfe

---

initialize $\boldsymbol{P}_{(0)}$
**for** $t = 0, 1, 2, \dots$ until convergence **do**
  $\boldsymbol{W}_d = \arg\min_{\boldsymbol{W}} -\mathrm{Tr}(\boldsymbol{W}_{(t)}^\dagger\boldsymbol{W}) + g(\boldsymbol{W}\boldsymbol{X})$
  $\alpha \leftarrow 1$
  **while** $-\log|\det(\boldsymbol{W}_{(t)} + \alpha_t(\boldsymbol{W}_d - \boldsymbol{W}_{(t)}))| > -\log|\det\boldsymbol{W}_{(t)}| + (\alpha/2)\mathrm{Tr}(\boldsymbol{W}_{(t)}^{-1}(\boldsymbol{W}_d - \boldsymbol{W}_{(t)}))$ **do**
    $\alpha \leftarrow \alpha/2$
  **end while**
  $\boldsymbol{W}_{(t+1)} = \boldsymbol{W}_{(t)} + \alpha(\boldsymbol{W}_d - \boldsymbol{W}_{(t)})$
**end for**

---

Regarding the line search step, the backtracking line search (Armijo rule) (Bertsekas, 1999) is used to guarantee sufficient decrease of the objective function. Since $g$ is convex, as long as $\boldsymbol{W}_{(t)}$ is feasible, then $\boldsymbol{W}_{(t+1)}$ is also feasible since it is a convex combination of $\boldsymbol{W}_{(t)}$ and $\boldsymbol{P}_d$, which are by definition feasible. Therefore, the only nontrivial part is to find a feasible initialization $\boldsymbol{P}_{(0)}$. This can be done by optimizing an arbitrary linear objective subject to $g$ (which means one should not apply line search at this step).

Frank-Wolfe was proposed to solve an instance of (1) by Hu & Huang on SCA (2023a) and NCA/ACA (2024).

The two proposed algorithms 2 and 1 shows striking similarities, especially considering they both fundamentally solve linear programs in each iteration. For nonconvex optimization, both algorithms guarantees that every limit point is a stationary point with some additional assumptions, which is reassuring to know. The differences are as follows:

1. Block coordinate descent is guaranteed to monotonically improve the objective function by design, so there is no need for line search as in Frank-Wolfe. This could save some computation as calculating $\det\boldsymbol{W}$ may not be cheap when $k$ is large.

2. On the other hand, each iteration of Frank-Wolfe only need to calculate $\boldsymbol{W}^\dagger$ once for the update of all its $k$ rows, whereas BCD requires to recalculate $\boldsymbol{W}^{-1}$ for each of its row updates. Overall, the per-iteration complexity is almost identical.

## A.3 NUMERICAL PERFORMANCES

We now provide some numerical experiments to demonstrate how our proposed L-ADMM performs compared to the existing algorithms outlined in this section. The settings are exactly the same as in §4, except that in the figures the horizontal axis shows time elapsed in seconds, not iterations—while L-ADMM takes hundreds to thousands of iterations to converge, neither BCD nor FW takes more than a few tens, but their per-iteration complexities are much higher as they require solving linear programs as a sub-routine.

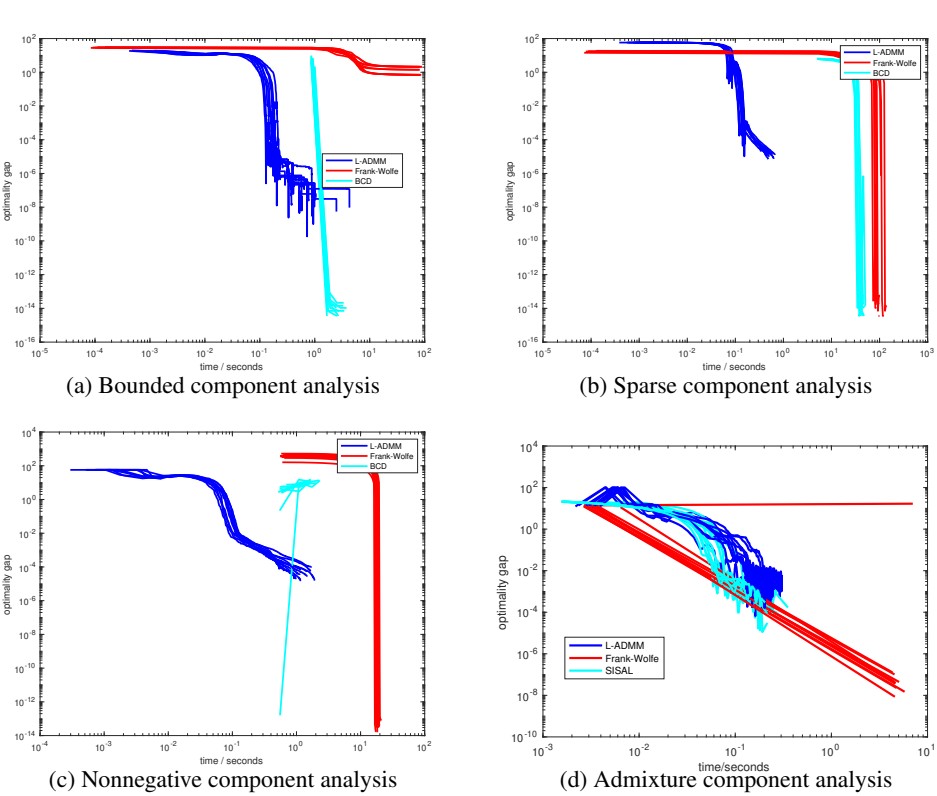

Figure 1: Convergence of L-ADMM vs. BCD and FW for various latent component analyses models on 10 random trials.

The convergence behavior of 10 random trials of L-ADMM, BCD, and FW are shown for BCA, SCA, and NCA in Figure 1. For ACA, we find that BCD proposed by Chan et al. (2009) takes too long to converge in all cases due to the fact that the constraints are not separable over the rows, so BCD is not included; instead, we include another augmented Lagrangian method called SISAL designed specifically for ACA as comparison. As we can see, L-ADMM works significantly better than BCD and FW in all cases. In some cases, BCD or FW may not even solve the problem, such as FW for BCA or BCD for NCA, which is in fact reasonable as we are trying to solve a nonconvex problem—what *is* incredible is that L-ADMM always works within a very short amount of time, which motivates this works. L-ADMM is overall comparable to the highly efficient SISAL, which is a dedicated algorithm for ACA, while L-ADMM is also highly flexible for other problems.

## B   PROOF OF LEMMA 3.2

Recall the proposed L-ADMM is

$$
\begin{cases}
\boldsymbol{W}_{t+1} \leftarrow (\boldsymbol{S}_t + \boldsymbol{U}_t + \gamma \boldsymbol{S}_t^{\dagger\top})\boldsymbol{X}^{\dagger} \\
\boldsymbol{S}_{t+1} \leftarrow \mathrm{Prox}_{\gamma g}(\boldsymbol{W}_{t+1}\boldsymbol{X} - \boldsymbol{U}_t) \\
\boldsymbol{U}_{t+1} \leftarrow \boldsymbol{U}_t + \boldsymbol{S}_{t+1} - \boldsymbol{W}_{t+1}\boldsymbol{X}
\end{cases}
\tag{2}
$$

and Lemma 3.2 is

**Lemma B.1.** *When running the L-ADMM iterations, if $\boldsymbol{S}_t$ satisfies*

$$
\log |\det \boldsymbol{S}_t \boldsymbol{S}_\natural^{\dagger}| \le \mathrm{Tr}\, \boldsymbol{S}_t \boldsymbol{S}_\natural^{\dagger} - k,
\tag{3}
$$

*then $\boldsymbol{S}_{t+1}$ also satisfies*

$$
\log |\det \boldsymbol{S}_{t+1} \boldsymbol{S}_\natural^{\dagger}| \le \mathrm{Tr}\, \boldsymbol{S}_{t+1} \boldsymbol{S}_\natural^{\dagger} - k,
$$

It is well known that $\mathrm{Tr}(\boldsymbol{S}_t \boldsymbol{S}_\natural^{\dagger})$ equals to the sum of its eigenvalues, and $\det(\boldsymbol{S}_t \boldsymbol{S}_\natural^{\dagger})$ equals to the product of its eigenvalues. Since we do not restrict $\boldsymbol{S}_t \boldsymbol{S}_\natural^{\dagger}$ to be symmetric, its eigenvalues may be complex. However, we do restrict $\boldsymbol{S}_t \boldsymbol{S}_\natural^{\dagger}$ to be real, so its complex eigenvalues always come in conjugate pairs. This means if a complex eigenvalue takes the form $\rho + i\eta$, then $\rho - i\eta$ is also an eigenvalue.

A sufficient condition for (3) is that all eigenvalues satisfy

$$
\rho - 1 \ge \frac{1}{2}\log(\rho^2 + \eta^2).
\tag{4}
$$

This inequality obviously holds if $\eta = 0$ (i.e., this eigenvalue is real) when $\rho > 0$ as per the famous inequality $\rho - 1 \ge \log \rho$. What is somewhat less obvious is that this inequality also holds when $\rho > |\eta|$ when $\eta \ne 0$. It is easy to verify that for a fixed $\eta$, the second derivative of $-(1/2)\log(\rho^2 + \eta^2)$ with respect to $\rho$ is positive when $\rho > |\eta|$. Thus invoking the first-order condition for a convex function at point 1, we obtain the inequality (4).

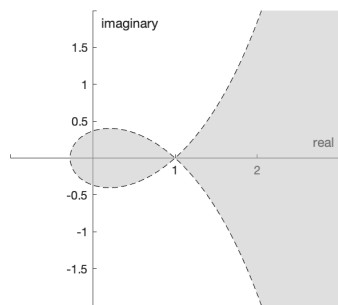

Figure 2: If all the eigenvalues of $\boldsymbol{S}_t \boldsymbol{S}_\natural^{\dagger}$ fall in the shaded region on the complex plane, then the inequality (3) is satisfied.

The resulting condition $\Re(\lambda) > |\Im(\lambda)|$ for every eigenvalue $\lambda$ of the matrix $\boldsymbol{S}_t \boldsymbol{S}_\natural^{\dagger}$ defines a region on the complex plane in which it can reside in order for (3) to hold. This region is illustrated in Figure 2. Note that if (4) is true for $\boldsymbol{S}_t \boldsymbol{S}_\natural^{\dagger}$, it is trivially true for its inverse $(\boldsymbol{S}_t \boldsymbol{S}_\natural^{\dagger})^{-1}$ as well.

*Proof of Lemma 3.2.* Replacing $\boldsymbol{X} = \boldsymbol{A}_\natural \boldsymbol{S}_\natural$ in (2), we first obtain from the $\boldsymbol{W}$ update that

$$
\boldsymbol{W}_{t+1}\boldsymbol{A}_\natural = (\boldsymbol{S}_t + \boldsymbol{U}_t)\boldsymbol{S}_\natural^{\dagger} + \gamma(\boldsymbol{S}_t \boldsymbol{S}_\natural^{\top})^{-\top},
$$

and the $\boldsymbol{U}$ update gives

$$
\boldsymbol{U}_{t+1}\boldsymbol{S}_\natural^{\dagger} = (\boldsymbol{U}_t + \boldsymbol{S}_{t+1})\boldsymbol{S}_\natural^{\dagger} - \boldsymbol{W}_{t+1}\boldsymbol{A}_\natural.
$$

Canceling $W_{t+1}A_\natural$ gives us

$$(S_{t+1} - U_{t+1})S_\natural^\dagger = S_t S_\natural^\dagger + \gamma(S_t S_\natural^\top)^{-\top}. \tag{5}$$

Since we assume $S_t S_\natural^\dagger$, so is $(S_t S_\natural^\top)^{-\top}$, and noticing that they share the same eigen basis, all eigenvalues of (5) satisfies (4), therefore we have

$$\log|\det(S_{t+1} - U_{t+1})S_\natural^\dagger| \le \mathrm{Tr}(S_{t+1} - U_{t+1})S_\natural^\dagger - k. \tag{6}$$

Notice that

$$S_{t+1} - U_{t+1} = W_{t+1}X - U_t,$$

while

$$S_{t+1} = \mathrm{Prox}_{\gamma g}(W_{t+1}X - U_t),$$

so we equivalently have

$$S_{t+1} = \mathrm{Prox}_{\gamma g}(S_{t+1} - U_{t+1}).$$

Since we have (6), by using the property of proximal operator, we have

$$\log|\det \mathrm{Prox}_{\gamma g}(S_{t+1} - U_{t+1})S_\natural^\dagger| \le \mathrm{Tr}\, \mathrm{Prox}_{\gamma g}(S_{t+1} - U_{t+1})S_\natural^\dagger - k.$$

This completes the proof. $\qquad\square$

## C  PCA EQUIVALENCE

Consider the main formulation with $g(\cdot) = (1/2)\|\cdot\|_\mathrm{F}^2$:

$$\underset{W}{\mathrm{minimize}} \quad -\frac{1}{2}\log\det WW^\top + \frac{1}{2}\|WX\|_\mathrm{F}^2. \tag{7}$$

Taking the gradient with respect to $W$ and setting it equal to zero at optimum $W_\star$, we have

$$W_\star^{\dagger\top} = W_\star XX^\top.$$

We assume that $W_\star$ is a wide matrix with linearly independent rows, so $W_\star W_\star^\dagger = I$, therefore multiplying both sides by $W_\star^\top$ on the right gets

$$I = W_\star XX^\top W_\star^\top$$

This shows two things:

1. The symmetric matrix $XX^\top$ can be diagonalized by $W_\star$. This can happen if rows of $W_\star$ are eigenvectors of $XX^\top$ and their Euclidean norms equal to one over the square root of their corresponding eigenvalue.

2. At optimum, the second term in (7) exactly equals to $(1/2)\|W_\star X\|_\mathrm{F}^2 = k/2$, meaning that we only need to worry about the first term when picking the eigenvectors of $XX^\top$ to construct $W_\star$.

As a result, it is easy to conclude that an optimal solution is $W_\star = \Sigma_k^{-1}U_k^\top$, where $\Sigma$ is a diagonal matrix with the $k$ largest singular values of $X$ on the diagonal, and columns of $U_k$ are their corresponding left singular vectors. Note that $QW_\star$ would also be optimal for any orthogonal matrix $Q$, showing that this model is not identifiable.

This indeed provides yet another interpretation of the famous PCA problem.