# OpenReview forum: "Provably Convergent Nonconvex Algorithm for Volume Optimization-based Latent Component Analyses"
_ICLR.cc/2026/Conference — Submitted to ICLR 2026_

### Official Review · Reviewer_H3Ks · 2025-10-28

**Soundness:** 2
**Presentation:** 2
**Contribution:** 3
**Rating:** 2
**Confidence:** 4

**Summary:**

This paper presents a unified optimization framework for several latent component analysis problems, backed by a novel L-ADMM algorithm with a claimed global convergence guarantee. The unification of these problems under a single volume-maximization principle is a valuable conceptual contribution. However, the validity of the central theoretical claim remains unverifiable due to the omission of a key proof, and the empirical validation is preliminary, lacking comparisons to established baselines and tests on real-world data. These issues currently preclude a full assessment of the method's practical impact.

**Strengths:**

1. The paper effectively unifies several distinct unsupervised learning models (BCA, SCA, NCA, ACA) under a single volume-optimization framework, offering a valuable high-level perspective.
2. Tackling the global convergence of non-convex problems like complete dictionary learning is a challenging and impactful goal. The proposed L-ADMM variant presents a novel technical pathway.

**Weaknesses:**

1. The global convergence guarantee (Theorem 3.1) relies entirely on Lemma 3.2. The proof of this foundational lemma is relegated to the supplement, making it impossible to assess the validity of the paper's central contribution.
2. The theoretical guarantee depends on an initialization condition (Eq. 17) that requires knowledge of the unknown ground-truth solution S_♮, rendering it hard to use in practice. The claim that this condition is easily met with random initialization lacks any statistical evidence.
3.  The empirical evidence is not enough. There are no comparisons to established baselines, validation is confined to idealized synthetic data, and there is no empirical scrutiny of the theoretical assumptions during iterations.
4. The paper's organization could be improved. Critical flaws include: (a) the lack of a dedicated "Related Work" section, failing to contextualize the novelty against prior art; (b) a disjointed introduction with a misplaced technical subsection (1.1); and (c) redundant content. They obscure the paper's narrative and contributions.

**Questions:**

1. The complete proof of Lemma 3.2 must be included in the main text to allow for verification of Theorem 3.1.
2. Can a practical initialization scheme not relying on S_♮ be proposed? Please provide statistical evidence (e.g., success rate over thousands of random trials) for the claim that condition (17) is easily satisfied.
3. It is essential to add comparisons to state-of-the-art baselines and include results on real-world datasets. An experiment monitoring the key theoretical inequality during iterations would be highly valuable.
4. The paper must be restructured. A dedicated "Related Work" section is essential. The technical content of Section 1.1 should be moved to a new section, etc.

---

### Official Review · Reviewer_5sLf · 2025-10-31

**Soundness:** 2
**Presentation:** 2
**Contribution:** 2
**Rating:** 2
**Confidence:** 4

**Summary:**

This paper studies a linearized ADMM variant for volume optimization-based latent component analyses. A convergence result is provided in the convex setting and the same result is extended to a family of volume optimization-based problems, provided the initialization is inside some basin of attraction. Some experiments suggest that the method can converge to the global minimum.

**Strengths:**

1. The linearization around $A^\dagger(c-Bs_t)$ is interesting and this choice is well-motivated by the given convergence proof in the convex setting.
2. The problem of interest (latent component analyses) is important to the machine learning community.

**Weaknesses:**

1. The initialization condition for convergence of the volume-based latent component analyses depends on the solution. This condition is claimed to be mild, yet I find the motivation lacking. What is missing is either (i) a thorough analysis of this initialization condition for some specific problems with known distributions of the solution or (ii) more convincing empirical evidence (which should also include the fraction of initializations that satisfied the condition) better showcasing how it behaves in practice.
2. Related to the previous weakness, it is not discussed in the experiments how the initializations were chosen. If this is taken from the same distribution as the true underlying solution, it might explain why the condition is ‘easy’ to satisfy.
3. The writing quality isn’t great and the figures are also unclear (see issues and questions below).
4. It is claimed that (4) in the supplementary material is satisfied for $\rho > |\eta|$. This is clearly not true by Figure 2 in the supplementary.

Minor issues/suggestions:
1. The conditions given for the identifiability for each of the four cases feels out of place in the main text, since it is not really relevant to the L-ADMM algorithm, which should be the main focus of the text.
2. Add dimensions to all matrices (at least in theorem statements) since it often matters whether they are square or rectangular. Also include the rank assumptions more explicitly since this is crucial for the properties of the pseudo inverse. For example line 354 requires $X$ to be full row rank, which I cannot find anywhere in the text.
3. Plot averages and error bars in figures instead of all trajectories.
4. Line 135: disk -> cell
5. Line 259 & 270: should refer to (8), not (5)
6. Line 308: = -> $\leq$
7. Line 317: orthogonizing -> orthogonalizing
8. Line 345: “the gradient of we know the gradient of the”
9. Line 382: a $\dagger$ seems to be missing on $S_t^\top$
10. Line 393: is -> its
11. Line 465: == -> =
12. Line 220 in the supplementary: “Since we assume $S_t S^\dagger_\natural$, so is $(S_tS_\natural^\top)^\top$” What is assumed?

**Questions:**

1. Provide more details for the simplification to obtain (16).
2. The experiments show the optimality gap going to 1e-6, yet other methods seem to go to lower values. How do you then know that the found solutions are global optima?
3. Regarding the figure in the supplementary material: why is time plotted on a log scale? why don’t all methods start at the same point? why does the optimality gap in (c) have a sudden jump upwards for the BCD method?
4. Please rewrite the proof of Lemma 2, which is a crucial part of the claimed core contribution of the paper, yet difficult to read. I am currently not certain about the correctness of this result, especially in light of the fourth stated weakness.
5. How are the least squares problems solved? Is the matrix prefactorized? If yes, is this time properly taken into account in the comparisons?

---

### Official Review · Reviewer_BFQE · 2025-11-01

**Soundness:** 3
**Presentation:** 3
**Contribution:** 3
**Rating:** 6
**Confidence:** 2

**Summary:**

This paper presents a novel algorithm for a family of nonconvex optimization problems common in unsupervised machine learning. The primary contribution is an algorithm based on Linearized Alternating Direction Method of Multipliers (L-ADMM) that is provably guaranteed to converge to a global optimum at a sublinear rate if the initialiaztion is good enough (lies in the basin of attraction), despite the problem's nonconvex and NP-hard nature.

**Strengths:**

The core contribution of the paper mainlly focus on the convergence analysis of equation $(4)$. A key technical contribution is the specific variant of L-ADMM used. Standard L-ADMM linearizes the difficult part of the objective ($f(w)$) at the previous iterate ($w_t$). This paper's variant, however, linearizes $f$ at a different point ($A^\dagger(c - Bs_t)$).

**Weaknesses:**

1) Though the convergence of the optimal Lagrangian function value is provided in theorem $2.1$, it is still unknown what is the convergence rate of the primal variable $w_\star$. I am wondering if the auhors could indicate the corresponding rate.

2) To the best of my knowledge, for general non-convex problem, ADMM like algorithm can only guarantee to converge to the stationary points. I am wondering if the authors could point out what is the key structures that helps to ensure the convergence to the global optimum in this case.

**Questions:**

1) The authors state that the initialization requirement in Equation $(16)$ differs from those of other non-convex algorithms. This claim is unclear and requires further elaboration. The authors should be more precise about this distinction and clarify the significance of the 'gap' between the requirements presented in this paper and those of existing methods."

2) In my understanding, the log-det term in the equation $1$ is used to orthogonalize the rows vectors of $W$ for which can be regarded as the penalty functions of constrains $ WW^T= I$. I am wondering what will the case be like if we replace the log-det term with something similar with $\|WW^T-I\|_F$.

---

### Official Review · Reviewer_KMXJ · 2025-11-01

**Soundness:** 2
**Presentation:** 2
**Contribution:** 1
**Rating:** 2
**Confidence:** 4

**Summary:**

This paper considers a unified framework for a family of latent component analysis (LCA) problems under the lens of volume optimization.
The authors formulate a general nonconvex optimization problem of minimizing the negative log-determinant (to maximize the “volume” of the mixing matrix) plus a regularization term that encodes task-specific constraints. To address this problem, they propose a Linearized ADMM algorithm and provide theoretical guarantees. Experiments on synthetic and real datasets suggest that the proposed method performs comparably or better than baseline algorithms across several LCA formulations.

**Strengths:**

1. This paper connects several LCA variants via a single “maximum volume” optimization formulation, revealing structural commonalities across bounded/sparse/nonnegative/admixture models. The geometric interpretation via log-det volume maximization and identifiability arguments provides some intuition.

2. The linearized ADMM updates rely on standard matrix operations, making the algorithm easy to implement.

3. Experiments demonstrate superior performance, suggesting that the proposed algorithm is effective in practice.

**Weaknesses:**

1. The convergence results presented in Theorems 2.1 and 3.1 are neither practical nor meaningful, as they rely on the assumption that the optimal dual multiplier $\lambda_*$ is known.

2. The ADMM and linearized ADMM algorithms discussed in this paper are not directly applicable to the LCA optimization models considered here, since they assume the smooth function $f(\cdot)$ is convex, whereas the smooth term $-\tfrac{1}{2}\log\det(WW^{\top})$ is non-convex.

3. The optimization problem in (1) is, in general, NP-hard. I remain unconvinced that this problem can be solved with guaranteed convergence to a global optimum.

**Questions:**

NA

---

### Meta-Review · Area_Chair_vMKt · 2026-01-18

**Summary:**

This paper proposes an algorithm based on a linearized ADMM for a family of nonconvex volume-maximization problems arising in latent component analysis, including bounded, sparse, nonnegative, and admixture models. The work aims to unify these problems under a single optimization framework and claims global convergence guarantees under certain initialization conditions.

Reviewers KMXJ, 5sLf, and H3Ks questioned the validity and practicality of the global convergence claims. The convergence analysis relies on assumptions that are difficult to satisfy in practice. In particular, the claimed “mild” initialization conditions are either poorly justified or unverifiable.

Reviewers 5sLf and H3Ks also pointed out weaknesses in the organization and writing, including the absence of a dedicated “Related Work” section, numerous typos, and redundant content.

Reviewer H3Ks further noted limited comparisons to established methods and a lack of evaluation on real-world datasets.

**Reviewer Concerns:**

The authors did not provide any response during the rebuttal period.

**Reviewer Scores:**

Since the authors did not provide any response during the rebuttal period, it is unlikely that any reviewer would have changed their original scores.

---

### Decision · Program_Chairs · 2026-01-26

Reject